# Optical Imaging in Human Lymph Node Specimens for Detecting Breast Cancer Metastases: A Review

**DOI:** 10.3390/cancers15225438

**Published:** 2023-11-16

**Authors:** Maria Papadoliopoulou, Maria Matiatou, Spyridon Koutsoumpos, Francesk Mulita, Panagiotis Giannios, Ioannis Margaris, Konstantinos Moutzouris, Nikolaos Arkadopoulos, Nikolaos V. Michalopoulos

**Affiliations:** 14th Department of Surgery, Attikon University Hospital, Medical School, National and Kapodistrian University of Athens, 1 Rimini Street, 12462 Athens, Greecenmichal@med.uoa.gr (N.V.M.); 2Laboratory of Electronic Devices and Materials, Department of Electrical & Electronic Engineering, University of West Attica, 12244 Egaleo, Greece; 3Department of Surgery, General University Hospital of Patras, 26504 Rio, Greece; 4Barcelona Institute of Science and Technology, Institute for Research in Biomedicine, IRB Barcelona, 08028 Barcelona, Spain; 51st Propaedeutic Department of Surgery, Hippocration General Hospital, Medical School, National and Kapodistrian University of Athens, 114 Vasilissis Sofias Avenue, 11527 Athens, Greece

**Keywords:** lymph node assessment, spectroscopic techniques, elastic scattering spectroscopy, optical coherence tomography, Raman spectroscopy, near-infrared fluorescence imaging, diffuse reflectance spectroscopy, Fourier-transform infrared spectroscopy

## Abstract

**Simple Summary:**

The evolution of optical imaging in identifying normal from malignant tissues offers great advantages in terms of speed and accuracy of diagnosis. Our results demonstrate that spectroscopic techniques can create informative and rapid tools that can be used to assess axillary lymph node status in breast cancer patients and guide further surgical decision-making.

**Abstract:**

Assessment of regional lymph node status in breast cancer is of important staging and prognostic value. Even though formal histological examination is the currently accepted standard of care, optical imaging techniques have shown promising results in disease diagnosis. In the present article, we review six spectroscopic techniques and focus on their use as alternative tools for breast cancer lymph node assessment. Elastic scattering spectroscopy (ESS) seems to offer a simple, cost-effective, and reproducible method for intraoperative diagnosis of breast cancer lymph node metastasis. Optical coherence tomography (OCT) provides high-resolution tissue scanning, along with a short data acquisition time. However, it is relatively costly and experimentally complex. Raman spectroscopy proves to be a highly accurate method for the identification of malignant axillary lymph nodes, and it has been further validated in the setting of head and neck cancers. Still, it remains time-consuming. Near-infrared fluorescence imaging (NIRF) and diffuse reflectance spectroscopy (DFS) are related to significant advantages, such as deep tissue penetration and efficiency. Fourier-transform infrared spectroscopy (FTIR) is a promising method but has significant drawbacks. Nonetheless, only anecdotal reports exist on their clinical use for cancerous lymph node detection. Our results indicate that optical imaging methods can create informative and rapid tools to effectively guide surgical decision-making.

## 1. Introduction

Breast cancer accounts for 2.3 million new cases per year and is the most common malignancy in the female population according to GLOBOCAN 2020 [1]. It is also the leading cause of malignancy-related mortality in females, accounting for 7.1% of all cancer-related deaths. However, breast carcinoma carries a favorable prognosis, and reported 5-year survival rates are between 76% and 90%, depending on stage and histology. Improvement of survival rates and years of life lost to disability is largely attributed to the betterment of diagnostic modalities, leading to earlier and more accurate diagnoses.

It is well-known that breast cancer spreads locoregionally through the lymphatic system. Hence, an assessment of regional lymph node status is useful in staging the disease and serves as a prognostic marker as well. The 5-year disease-free survival (DFS) rate decreases from 98% for node-negative patients to approximately 94% for patients with axillary lymph node macrometastases [2,3].

### 1.1. Histopathological Assessment

Lymph node status was historically assessed by axillary lymph node dissection (ALND), a process that involves dissection and removal of the axillary lymph nodes and subsequent histological examination [4]. However, this technique has been related to severe complications, including lymphedema, leading to significant discomfort, long-term impairment, or even handicap. During the last decades, sentinel lymph node biopsy (SLNB) has become the standard of care, replacing ALND for staging of the axilla. SLNB is a method to identify, and, therefore, determine the status of the first (sentinel) lymph nodes receiving lymphatic drainage from the primary tumor. Numerous techniques have been described for the accurate identification of sentinel lymph nodes, using either single tracing (dye or radioisotope) or double tracing modalities. A double technique, by injecting both a technetium-labeled nanocolloid and a blue dye around the tumor bed or periareolarly, has been shown to be among the most accurate and most widely adopted in everyday surgical practice. Reportedly, the identification rate of the double technique is as high as 96%, with a false-negative rate of 7.3% [4]. Possible drawbacks can be the unavailability of the isotope and severe allergic reactions related to the injected dye substances, as well as the requirement for formal histopathological examination of the excised lymph nodes and associated elevated costs. Even though SLNB has proven to be more effective in terms of quality of life and less costly, when compared to ALND, it is still associated with a significant financial burden of up to USD 4206 per patient, including additional length-of-stay costs [5]. Additionally, local resources and appropriate expertise need to be considered [4,6]. 

### 1.2. Imaging Techniques

Preoperative assessment of lymph node status with the use of existing imaging techniques has shown variable accuracy [6]. For ultrasound, the reported sensitivity and specificity are between 25 and 60% and 70 and 100%, respectively. For fluorodeoxyglucose (FDG)-positron emission tomography (PET), sensitivity ranges between 37% and 85%, while specificity is around 84–100%, making both techniques less reliable than routine SLNB biopsy that is nowadays recommended for all breast malignancies, irrespective of imaging results. Lastly, the respective values for contrast-enhanced magnetic resonance imaging (MRI) are 63–100% and 56–100%. Alternative methods, such as indocyanine green fluorescence (ICG), contrast-enhanced ultrasound (CEUS) using microbubbles, superparamagnetic iron oxide nanoparticles (SPIO), and others, have also been used to assess lymph node status, albeit with reportedly low sensitivities [4,7].

### 1.3. Basic Principles of Light–Tissue Interaction and Optical Diagnostic Methods

An emerging field that is closely related to cancer research is optical diagnostic methods. Optical imaging uses light properties to obtain detailed images of organs, tissues, cells, and even molecules [8,9,10]. Assessment of specimens using the herein-described techniques is an actively investigated topic, showing promising results in early disease detection, diagnosis, and treatment response monitoring. The scope of this work is to present and further elucidate the use of less-discussed spectroscopic imaging modalities in the detection of lymphatic spread in breast cancer patients. A summary of the framework of spectroscopic imaging on axillary lymph nodes is shown in Figure 1.

Before delving further into the world of electromagnetic imaging in lymphatic mapping for breast cancer, it is important to establish the biomechanics behind the images. The interaction of electromagnetic radiation with biological tissues gives rise to several optical effects that are highly relevant to diagnostic applications; they include light reflection, scattering, and absorption [8,9] (Figure 2). The evolution of these effects depends on the optical properties of the irradiated medium, which can be quantified via a set of optical constants, the most important of which are the refractive index and the coefficients of scattering, absorption, and anisotropy [10,11,12,13,14,15,16,17]. It is worth noting that these optical constants exhibit strong dependencies on the wavelength of light, as well as spatial variations that reflect tissue composition and inhomogeneities on macro, meso, and microscopic scales.

Light reflection occurs at the boundary separating two regions of different refractive indices, such as the interface between a tissue sample and its surrounding medium (e.g., air). Generally, two distinctive types of reflection coexist, known as specular and diffused reflection. Specular reflection describes light rays bouncing off the interface at an angle that is equal to the angle of incidence in a mirror-like geometry. Diffused reflection describes light rays deflecting in arbitrary directions within the entire hemisphere adjacent to the interface. A special case of diffused reflection is the so-called back-reflection, which refers to the deflected wave that follows the path of the incidence beam, toward the light source. The standard mathematical treatment of reflection follows the Fresnel theory for its specular component and the Kubelka–Munk theory for its diffused one [18,19].

Light scattering may be perceived as the change in the propagation direction of photons due to collisions with other quantum particles (“scatterers”). Scattering can be either (a) elastic, in which case the photon energy (equivalently, the light frequency) remains unaffected by the collision, or (b) inelastic, in which case part of the colliding particle’s energy is transferred from one to another, causing either a downshift or an upshift in photon frequency, which is known as the Stokes shift and anti-Stokes shift, respectively. In biological tissue, scatterers vary in size, shape, and form as a result of the complex organization at different scales; they may include, for example, fiber structures, cells, nuclei, and organelles, in the case of elastic interactions, or quanta associated with the vibrational states of molecular bonds in the case of inelastic interactions. The analytical treatment of elastic scattering is based on the Mie and Rayleigh theories, which apply to situations where the scatterers are larger or comparable in size to the light wavelength, respectively. The predominant type of inelastic scattering in biological matter follows the Raman theory [20,21,22,23,24,25].

Light absorption is the process by which photons become extinct, depositing their energy to absorption centers (typically, atomic, or molecular electrons), inducing subsequent photochemical and photothermal effects [8]. Common absorbers in biological tissues include water, lipids, and proteins. Various spectroscopic methods (cumulatively known as “absorption spectroscopy”) rely on the fact that each absorber has unique spectral characteristics, which serve as molecular signatures [26,27,28]. Fluorescence spectroscopy/imaging is a second class of diagnostic methods that also rely on light absorption by certain absorbers that exhibit fluorescent properties (“fluorophores”). Fluorescence refers to the excitation of a molecule to a higher energy state due to the absorption of a photon, followed by the re-emission of a lower energy photon, during the cascaded decay to the ground state. Two types of fluorescent signals are often exploited within this context [26,27]. The first is auto-fluorescence, which originates from several endogenous biomolecules, such as hemoglobin, typically having absorption/emission spectra in the ultraviolet/visible (UV/VIS) spectral range [26]. The second is labeled fluorescence, originating from an exogenous fluorophore, such as a dye, which selectively binds to a functional group of the targeted molecule [27].

Specular and diffused reflection reduces the amount of light from an external source that can be coupled into the tissue specimen. Scattering and absorption reduce the number of photons that travel ballistically (that is, in a straight line) within tissue; equivalently, these two effects regulate the penetration depth of a light beam into an inhomogeneous biological sample. There are three infrared spectral regions where the penetration depth maximizes (i.e., scattering and absorption reduction): the so-called first, second, and third “window”, extending from 650 nm to 900 nm, 1100 nm to 1350 nm, and 1600 nm to 1870 nm, respectively [28].

Furthermore, reflection, scattering, and absorption/fluorescence serve as signals in a rich variety of optical technologies that enable not only the visualization of tissue structures but also metabolic processes that take place at the sub-cellular lever. When this is translated to a medical image, it aids in providing functional information about different types of tissues, thus highlighting important differences between surrounding healthy tissue and cancerous lesions and improving the speed and accuracy of the diagnosis [28,29,30,31,32].

In the present article, we review six spectroscopic/imaging techniques that have been used in assessing human lymph nodes in breast cancer, namely: elastic scattering spectroscopy (ESS), optical coherence tomography (OCT), Raman spectroscopy, near-infrared fluorescence imaging (NIRF), diffuse reflectance spectroscopy (DRS), and Fourier-transform infrared spectroscopy (FTIR). Other spectroscopic techniques using fluorescent tracers to detect malignant lymph nodes (photoacoustic imaging—PAI) are also described in the literature with promising results but currently have been used only in animal models, so they are not within scope of the present article [33,34,35].

## 2. Materials and Methods

We conducted a narrative review using studies of various complexity and design. An electronic search of PubMed/MEDLINE and Scopus was carried out between March 2023 and June 2023, reviewing all the medical literature published in the English language regarding the use of optical imaging in the assessment of cancerous human lymph nodes, with a special interest in breast cancer lymph node assessment. We used a combination of the terms “Spectroscopic techniques” OR “Imaging techniques” OR “Optical properties” AND “Cancer” OR “Breast cancer” OR “Node assessment”. After removing duplicates, articles were screened for inclusion by reviewing their related abstracts and full texts. A manual search of the reference lists from the selected articles was also performed, identifying further articles that could possibly help contextualize our findings. All articles screened by the research team were put through Covidence, where further abstract and full-text screening took place, until the final list of included articles was agreed upon by all authors. Thirty-four articles were identified to fulfill the inclusion criteria. One article reviewed all mentioned techniques and another manuscript had information about optical coherence tomography and Raman spectroscopy. More specifically, our search found eight articles about elastic scattering spectroscopy, ten about optical coherence tomography, ten about Raman spectroscopy, four about near-infrared fluorescence imaging, three about diffuse reflectance spectroscopy, and two about Fourier-transform infrared spectroscopy.

Search protocol, article selection, and data extraction were assessed by two independent researchers. All disputes regarding the inclusion of manuscripts within our review were resolved by a case-based discussion between the two most senior researchers.

## 3. Results

A summary of the examined spectroscopic/ imaging techniques is presented in Table 1. 

### 3.1. Elastic Scattering Spectroscopy

Elastic scattering spectroscopy is an optical method that has been extensively used for diagnostic purposes. It is based on the detection of the elastically back-scattered light as a function of either wavelength or angle, which contains information on the scatterer’s size, shape, and density, thus relating to tissue pathology [43]. It is especially useful to determine the size, density, and status of important intracellular structures, such as nuclei density, nucleoli, and mitochondrial status, all of which constitute indicators of malignancy or invasive process, and can, therefore, improve information regarding the status of imaged tissues [44].

ESS is a highly reproducible method, requiring simple and inexpensive experimental components (i.e., a flash lamp is typically used as the pump source). A possible drawback is the need to distinguish, at the detector stage, single-scattered from multiple-scattered photons, which is a non-trivial algorithmic task. ESS is a process that has largely involved machine learning algorithms, such as Bayesian and Markov field models, for the determination of malignancy. ESS has been reported to be automated and successfully detect almost 70% of clinically relevant lymph nodes containing metastatic lesions [45]. However, ESS is not ideal for the examination of relatively small specimens and, hence, it may fail to identify small cancer metastases [6].

Jonhson et al. examined multiple spectra from 139 excised nodes of 68 breast cancer patients using ESS. Their method was able to detect the spectra from cancerous nodes with a reported sensitivity of 84% and a specificity of 91% [46]. Keshtgar et al. used elastic scattering spectroscopy for intraoperative diagnosis of breast cancer node metastases [47]. The researchers examined a total of 361 lymph nodes and created a diagnostic algorithm. They also obtained and analyzed scans from 129 independent lymph nodes. Macrometastatic disease (larger than 2 mm) was detected with a sensitivity of 76% and a specificity of 96%. 

Finally, elastic scattering spectroscopy has been investigated with promising results in the setting of lymph node metastases from oral cancer and prostate cancer, indicating the method is efficient and reproducible in other forms of cancer [36,48,49].

### 3.2. Optical Coherence Tomography

Optical coherence tomography can be perceived as a subset of ESS, exploiting the use of back-scattered light to produce cross-sectional two-dimensional (2D) or three-dimensional (3D) images by performing multi-axial measurements of time delay in the light echo [36]. For several decades, OCT has been employed in biomedical diagnosis, particularly in ophthalmology for the assessment of retinal and anterior chamber disease [37,50,51]. 

OCT’s main advantages include a high resolution, making micrometastatic scanning possible, a short data acquisition time compared to other tomographic approaches (typically, a few seconds are sufficient), as well as its potential to be used through a medical needle probe [4,37,50,51]. On the negative side, OCT is a relatively costly and experimentally complex method since it is based on the indirect measurement of the time delay by means of low-coherence interferometry. In other words, the elastically scattered light is not directly read by the detector/spectrometer modality, as in the case of standard ESS; instead, it interferes with a reference signal, whose path length is scanned in a two-arm Mickelson configuration [52]. Subsequently, heterodyne detection of the interference pattern enables the localization of scattering sites within the tissue sample. Therefore, pumping by incoherent light sources is not possible, but rather broadband coherent sources are required, such as short pulse lasers. As with the previously discussed modalities, OCT also offers the chance to incorporate machine learning into the diagnostic process, thus aiding in reducing operator workload and stress and shortening return times. In a 2018 study, automated OCT diagnosis had an overall accuracy of 90% in identifying excised lymph nodes [38]. The latter, along with the portability and ease of access to OCT machines, are strong arguments for the more widespread use of OCT, even in the intraoperative assessment of specimens.

Experimental applications of OCT have shown interesting results. Mclaughlin et al., in order to distinguish non-cancerous from malignant tissues, examined 30 lymph nodes and designed an algorithm to diminish the variations of OCT images due to location [51]. Their ex vivo study provided us with the first published report of characteristic OCT backscattering patterns in malignant human axillary lymph nodes. Nguyen et al. published a preliminary clinical study presenting the lymph node assessment from 17 patients with breast cancer. Scattering changes in the cortex, relative to the capsule, were used to differentiate between normal, reactive, and metastatic nodes [53]. In another study by Scolaro et al., parametric images of human axillary lymph nodes were obtained, by measuring local attenuation coefficients [54]. More recently, Nolan et al. used a portable intraoperative OCT system to evaluate labeled sentinel lymph nodes prior to formal histopathological examination [55]. Three-dimensional OCT datasets were recorded from one or more locations per lymph node in a process that lasted 5 to 10 min. Following a decision tree diagram, abnormal lymph nodes were identified based on scattering and heterogeneity. The above-mentioned method was reported to have a sensitivity of 58.8% and a specificity of 81.4%.

Overall, OCT seems to be a promising tool, capable of rapidly identifying lymph node metastases. As of now, existing studies on OCT seem to be based on single-institution, small-scale cohorts, making the use of a complicated issue requiring more experimental data on hand more widespread.

### 3.3. Raman Spectroscopy

Raman spectroscopy is a non-distractive fingerprinting tool with extensive applications in analytical chemistry, pharmacology, and medical diagnosis [55,56]. The operating principle of Raman spectroscopy is based on the detection of the wavelength shift during the inelastic interaction of photons (typically generated from a narrowband laser source) with the bonds of biomolecules, such as nucleic acids, proteins, and lipids [56].

Raman spectroscopy offers the advantage of extreme sensitivity, even to small changes in the sample structure. It is a sophisticated modality that can differentiate even minor changes in the biochemical structure of carbohydrates, lipids, and proteins, which indicate malignant transformation [57]. Nonetheless, this optical method depends on the detection of very weak signals (usually, the Stokes shift, which is stronger than the anti-Stokes shift), and requires relatively expensive instrumentation, such as lock-in detection for noise reduction. Moreover, Raman spectroscopy requires prolonged data acquisition time (typically, several minutes per spectrum), which can be particularly problematic when it comes to biomedical applications [4,39,58]. A prime example utilized in breast cancer specimen diagnosis is the accurate and easy measurement of DNA and RNA density, an increase in which is characteristic of uncontrolled cellular proliferation, which is a hallmark of cancer.

Horsnell et al. used Raman spectroscopy to examine 59 axillary lymph nodes in patients with breast cancer with a 5-point probe model and a 10-point probe model [40]. The sensitivity of the method was reported to be 81% (10-point probe model), with a corresponding specificity of 97% (both probes). Raman spectroscopy has also been investigated in the context of head and neck malignancies, providing high accuracy for the identification of malignant lymph nodes [41,59,60]. In a study by Lloyd et al., 103 head and neck suspicious lymph nodes were examined [60]. Models were developed to distinguish between benign, reactive, primary malignant, and secondary malignant nodes, with a reported sensitivity and specificity of 78–90% and 86–89%, respectively. Raman spectroscopy has also been used for the determination of surgical margins (cancerous infiltration versus free margins) in excised surgical specimens. An overall accuracy of 93% was achieved using a dispersive Raman spectrometer [42]. Another exciting application of Raman spectroscopy in breast cancer is the development of a handheld modality, the so-called Spectropen^®^, which can record fluorescence and Raman signals at the same time and is portable enough to be used in the operating room, directly within the SLNB cavity or the tumor cavity/specimens after breast-conserving surgery [61].

In conclusion, Raman spectroscopy creates a “fingerprint” by analyzing the chemical composition and molecular structure of a sample. Identification of cancerous lymph nodes seems to be feasible and accurate. Still, the method needs to be refined to be less time-consuming and facilitate intraoperative diagnosis [39].

### 3.4. Near-Infrared Fluorescence Imaging

Fluorescence imaging is based on the excitation and subsequent detection of either tissues’ auto-fluorescence or labeled fluorescence. It is a very well-established method for time-efficient and accurate biomedical diagnosis, including the ability to detect micrometastases [6,62,63].

The development of fluorophores with characteristic excitation and fluorescence spectra in the near-infrared region has enabled the operation of fluorescence spectroscopy beyond ultraviolet (UV) and visible wavelengths. Several fluorescent dyes have been approved for use in clinical applications, such as indocyanine green (ICG) and methylene blue [62]. Near-infrared fluorescence spectroscopy comes with two distinctive advantages. Firstly, near-infrared light penetrates deeper within the tissues. Secondly, filtering out the tissues’ ultraviolet/visible auto-fluorescence emission is a relatively simple method, which can reduce unnecessary noise to the measurements. 

Limited data exist from studies assessing NIRF in human lymph nodes for cancer detection, as the method has mostly been used in sentinel lymph node mapping [31]. Xia et al. carried out a clinical trial including four patients with prostate cancer and suspicious pelvic lymph nodes [63]. Intravenous ICG infusion was administered before surgery. Intraoperatively, an NIR imaging camera was used to identify suspicious lymphadenopathy. Three out of four preoperative lympadenopathies were identified as fluorescence-positive during in vivo examination. Ex vivo back-table examination subsequently identified all lympadenopathies as fluorescent-positive, while histologic examination confirmed the presence of metastatic deposits in all cases. Despite encouraging results indicating the possible use of near-infrared fluorescence imaging to guide pelvic lymphadenectomy, well-designed studies with larger samples are needed to draw safe conclusions.

### 3.5. Diffuse Reflectance Spectroscopy

Diffuse reflectance spectroscopy is a low-cost and easy-to-use tool for the assessment of tissue properties. It relies on the observation of different features appearing on the broadband spectrum that illuminates the sample compared to the spectrum of the diffusely reflected light [6]. Such differences convey information on the optical properties of the irradiated area (primarily, the scattering and the absorption coefficient) which, in turn, directly relate to physiological characteristics, such as tissue architecture, cellular density, neovascularization, and metabolic activity.

Experimental simplicity and cost efficiency of DRS are basically related to two facts. First, a single broadband lamp (typically emitting continuously in the ultraviolet/visible and near-infrared wavelengths) can be used as the pump source, avoiding the need for technologically complex and expensive broadband lasers. Second, exploitation of the back-reflection geometry enables the use of a single optical fiber (adaptable to small probes) to both deliver the pump light and collect the reflected signal. 

Kanick et al. developed a diagnostic algorithm to evaluate mediastinal lymph nodes from patients with lung cancer, and they subsequently devised an endoscopic probe carrying a diffuse reflectance spectroscopy system [64,65]. However, DRS has not been investigated in the setting of breast cancer axillary lymph nodes and seems to lack further experimental verification. The method appears to have been overtaken by other imaging techniques.

### 3.6. Fourier-Transform Infrared Spectroscopy

Fourier-transform infrared spectroscopy (FTIR) is an absorption technique based on the vibrational characteristics of chemical functional groups in relation to structure and composition [66]. Chemical functional groups can result in specific absorption peaks for the medium and thus lead to possibly identifying classes of molecules [6]. Typically, FTIR operates in a wavelength range of 250–2500 nm and has a relatively low penetration depth. The limitation of the technique is a strong absorption peak of water that implies a drawback in the examination of fresh tissue [6,66]. There are limited data on the applications of FTIR in the detection of malignant lymph nodes in humans.

Limited data are available on the application of FTIR for the detection of malignant lymph nodes in humans. Bird et al., in a study conducted in 2008, analyzed up to 100.000 spectra from 30 excised axillary lymph nodes that were also cross-examined by standard histological evaluation. Their results formed a diagnostic algorithm that could be used as an automatic unsupervised method to classify benign from malignant lymph nodes. This method had high reproducibility with both deparaffinized and frozen tissues [67]. In another study by Tian et al., 149 freshly removed SLNs from patients with breast cancer were characterized using Fourier spectroscopy. FTIR spectra were compared to results from histopathological evaluation and showed a sensitivity, specificity, and accuracy of 94.7%, 90.1%, and 91.3%, respectively [66]. Overall, based on results from the existing literature, Fourier-transform infrared spectroscopy can be exploited to develop an accurate and fast cost-effective method for the identification of malignancy of axillary lymph nodes. Nonetheless, further studies are warranted to improve the evidence base.

Other promising applications of FTIR that have been recently investigated are the detection of melanoma metastases and lymph node metastases in thyroid and gastric cancer. FTIR imaging of the primary tumors in bladder, thyroid, and colon cancer has also been shown to predict metastatic spread [68,69,70,71,72,73].

## 4. Limitations

Our study has an inherent limitation as it is a narrative study rather than a systematic review that includes five different imaging techniques. Still, this study was carried out in accordance with practice standards mentioned in the Methods section, which increases its scientific value. Most of the studies examined here suffer from low patient accumulation, largely due to their less than widespread use, and, therefore, their results are not ideally validated but rather constitute promising endeavors in the field of lymphatic imaging in breast carcinomas. Most of the techniques reviewed in the present study may suffer from limited reproducibility; measurements can differ due to altered conditions of the tissue–sensor interface and the environment’s circumstances, such as temperature and humidity.

Another significant limitation is the tissue/sample heterogeneity that seems to influence the technique. Importantly, published studies examined metastatic lymph nodes stemming from a variety of primary malignant neoplasms. Furthermore, few studies comment on potential outcome differences among discrete breast cancer histological subtypes. It is well-known that different histological subtypes have different affinities for lymphatic spread, metastasis, and mictometastases, and future efforts need to include sub-classification of the modalities’ performance in each different histological breast subtype. Moreover, age, race, or even menstrual phase can provoke significant tissue variations; for example, altering water and lipid content, which could possibly interfere with the method’s measurements and results [10].

## 5. Conclusions

Optical imaging has the potential to differentiate between normal and malignant tissues, offering great advantages in terms of speed and diagnostic accuracy. Our review indicates that optical imaging methods can create informative and rapid tools, which could be extremely useful in intraoperative lymph node assessment and surgical decision-making. Additionally, structural and sub-cellular data obtained by optical scattering and reflectance technologies can be combined with other optical modalities to characterize cellular state and structure–function relationships at a speed that is otherwise unachievable with the conventional microscopic pathological examination of excised tissue specimens. Further studies are needed to assess the in vivo intraoperative application of the described methods.

## Figures and Tables

**Figure 1 cancers-15-05438-f001:**
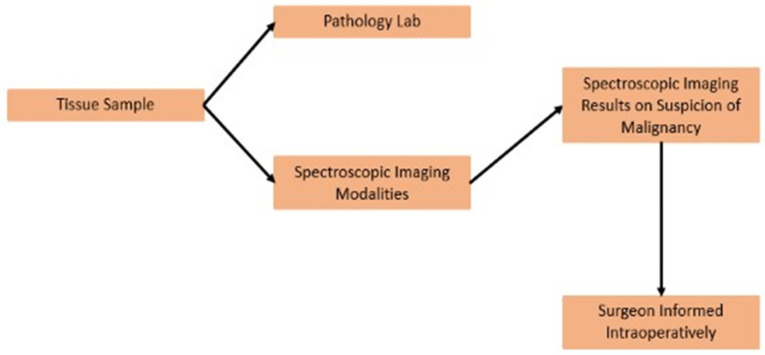
Working framework of spectroscopic imaging on axillary lymph nodes.

**Figure 2 cancers-15-05438-f002:**
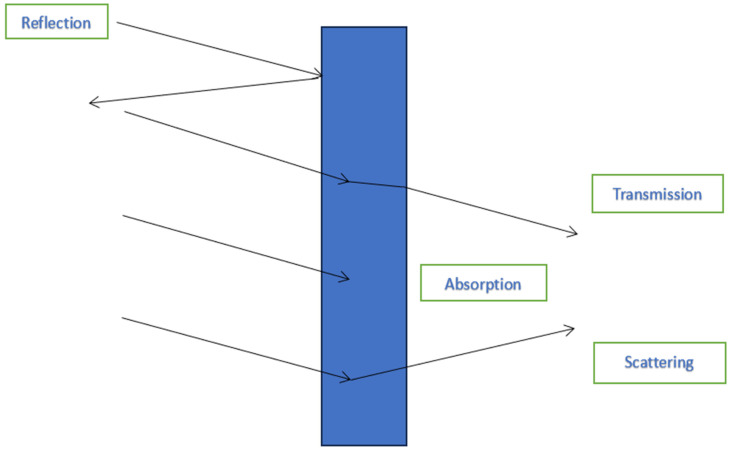
Basic principles of light–tissue interaction.

**Table 1 cancers-15-05438-t001:** Characteristics of spectroscopic/imaging techniques.

Technique	Cost	Depth	Resolution	Sensitivity/Specificity	Timing	References
ESS	low	0.5 mm	320–920 nm	76–84%/91–96%	20–25 min	[34,35]
OCT	high	2–3 mm	12 μm	58.8%/81.4%	real time	[4,36,37]
Raman	high	5–10 mm	10 μm	71–81%/97%	10–20 min	[4,38]
NIRF	low	5–10 mm	21–337 μm	limited data	real time	[39,40]
DRS	low	2–3 mm	200–1000 nm	limited data	real time to minutes	[4,41]
FTIR	low	2–3 mm	250–2500 nm	94.7%/90.1%	2–3 min	[4,42]

## Data Availability

The data presented in this study are available on request from the corresponding author. The data are not publicly available due to intellectual property.

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
