# Peer review of "Optical Imaging in Human Lymph Node Specimens for Detecting Breast Cancer Metastases: A Review"

_cancers, 2023, doi:10.3390/cancers15225438_

Round 1

Reviewer 1 Report

Comments and Suggestions for Authors

Undoubtedly, assessing the condition of regional lymph nodes in breast cancer has important staging and prognostic significance. Although formal histological examination is the current standard of care, optical imaging techniques have shown promising results in the diagnosis of diseases. In this article, the authors review five spectroscopic techniques, with particular emphasis on their use as an alternative tool for the evaluation of breast cancer lymph nodes.

1. We need a small section devoted to methodological details, how exactly the study is carried out, with a detailed description of the procedure, its advantages and limitations.

2. For each method, it is necessary to provide a table summarizing the existing literature data with a description of the groups studied so that the validity of the data obtained can be assessed.

3. Are the data in Table 1 on sensitivity and specificity averaged? or is this data from one of the authors?

4. Is infrared spectroscopy not used for the described tasks?

Author Response

Dear Reviewer,

We would like to thank you for taking the time to review our work on Optical imaging in human lymph node specimens for detecting breast cancer metastases. It is our belief that your insightful observations on our work will lead to the publication of the best version of our manuscript possible. The entirety of your suggestions is now incorporated within the manuscript. Please see in detail below:

1) The methodological section of our article has been enriched as suggested

2) A new table has been created to summarize the individual study results for each optical method

3) The data are averaged, based on the included studies. The respective references have also been provided

4) Your comment about infrared spectroscopy is accurate. It is an optical imaging technique that has been tested in the setting of breast cancer lymph node metastasis, which was unintentionally neglected in our manuscript. The existing literature has been revisited and care has been taken to include all available data in our revised manuscript

Please also find attached the revised manuscript, inclusive of the above-mentioned changes. Once again, we thank you for your consideration and time.

Kind regards,

The authors

Reviewer 2 Report

Comments and Suggestions for Authors

The article is a review on detection of lymph nodes metastasis with the use of various opical techniques, and it focus on human reserch. The research can support a decision of axillary lymph node dissection. The article is interesting and give detailed physical background, however, it needs some comments:

Introduction: Could you describe the difference in radiotherapy doses and extension, and 5-year survival with positive and negative sentinel node. What is the incidence of breast cancer metastasis to lymph nodes and other organs?

Lines 77-79: Isn't SPIO a kind of MRI contrast? Please, give more information of SPIO application.

Line 82: Give a definition of "the opical imaging".

Figure 2: The light should be refracted in both surfaces in transmission scheme.

Line 181: Give a number of the articles fulfilling the inclussion criteria (detailed according optical technique).

Figure 1: Cost of equipement or examination of a lymph node? Can you give the reference and quantity, please.

Line 270: Usually Raman spectroscopy research is performed using Stokes scattering which is stronger than ant-Stokes - can you comment that? Is it necessery to image whole lymph nodule with 10 um resolution? How long takes collecting singular Raman spectra? Are there menthods enhancing Raman intensity?

Dicussion: Describe the term of "heterogeneity". Is heterogeneity "the one entity". Are mollecular changes corellated with distribution of the signal of particular optical technique? Does distribution of the signal depand on the stage and cancer type? Can any opical imaging predict cancer metastasis to lymph nodes based on spectra of primary tomour? Is the signal of primary and secondary tumours similar? Is the intraoperative measurement better than the measurement before surgery? Is infrared spectroscopy useful for prediction of breast cancer metastasis to lymph node and organs in human or animal model?

Comments on the Quality of English Language

Line 67: What mean "local resources" in the sentence?

Line 87: "cancer patients.A summary " - add the space, please.

Lines 151, 152 - Add spaces between numbers and sizes (f.e. "from 650nm το 900nm" => from 650 nm to 900 nm).

Line 237: "nodes[41]." - add the space, please.

Line 289: "Spectropen®that" - add the space, please.

Author Response

Dear Reviewer,

We would like to thank you for taking the time to review our work on Optical imaging in human lymph node specimens for detecting breast cancer metastases. It is our belief that your insightful observations on our work will lead to the publication of the best version of our manuscript possible. The entirety of your suggestions is now incorporated within the manuscript. Please see in detail below:

1) We have expanded the introduction section of our manuscript to incorporate your comments

2) In lines 77-79 additional information about the SPIO technique has been included

3) In line 82 a formal definition of optical imaging is provided

4) Figure 2 has been appropriately amended

5) Additional information about the number of articles included per optical method has been added

6) In Figure 1 we have included data on time and average cost

7) Raman spectroscopy depends on the detection of very weak signals (usually, the Stokes shift, which is stronger than the anti-Stokes shift), and requires prolonged data acquisition time (typically, several minutes per spectrum), which can be particularly problematic when it comes to biomedical applications. In the experimental setup, the whole lymph node is examined so it is time-consuming.

8) Basic principles of the spectroscopic techniques are mentioned in 1.3. section (Basic principles of light-tissue interaction and optical diagnostic methods), so we avoided describing again those terms, in order to avoid repetition. To the best of our knowledge, there is no evidence of spectral differences between primary and secondary tumors and no comparative studies have been carried out to examine the difference between preoperative versus intraoperative measurements. Your point is accurate and we have reviewed the available literature on infrared spectroscopy and updated the manuscript accordingly. Please also find attached the revised manuscript, inclusive of the above-mentioned changes. Once again, we thank you for your consideration and time.

Kind regards,

The authors

Round 2

Reviewer 1 Report

Comments and Suggestions for Authors

I have no further comments on the article. I believe that in its present form the manuscript can be recommended for publication.

Author Response

Dear Reviewer,

We would like to thank you for taking the time to review our work on Optical imaging in human lymph node specimens for detecting breast cancer metastases. It is our belief that your insightful observations on our work will lead to the publication of the best version of our manuscript possible.

Please also find attached the revised manuscript with the last changes. Once again, we thank you for your consideration and time.

Kind regards,

The authors

Reviewer 2 Report

Comments and Suggestions for Authors

The article is suitable corrected and it gives insight into various optic techniques used in assessment in detection of breast cancer metastasis to lymph nodes, however:

- the Figure 2 needs correction (both surfaces a/b and b/a refract the light in tranmission, so the arrow should be double broken);

- I found more current reference for SPIO, line 86 (doi: 10.1245/s10434-023-13252-6);

- the tracers (imaging)/sensitizers (treatment) can be used for optoacoustic imaging and you can discuss it, if you see it fits herein (3 references: doi: 10.1038/s41377-020-00399-0, DOI: 10.1002/anie.201913149,  https://doi.org/10.1021/acsami.8b09142).

Author Response

Dear Reviewer,

We would like to thank you for taking the time to review our work on Optical imaging in human lymph node specimens for detecting breast cancer metastases. It is our belief that your insightful observations on our work will lead to the publication of the best version of our manuscript possible. The entirety of your suggestions is now incorporated within the manuscript. Please see in detail below:

1) Figure 2 has been appropriately amended

2) In line 86 the proposed reference about the SPIO technique has been included

3) Additional information about optoacoustic imaging with the proposed references was mentioned.

Please also find attached the revised manuscript, inclusive of the above-mentioned changes. Once again, we thank you for your consideration and time.

Kind regards,

The authors